# Efficacy of Intense Pulsed Light Combined Blood Extract Eye Drops for Treatment of Nociceptive Pain in Dry Eye Patients

**DOI:** 10.3390/jcm11051312

**Published:** 2022-02-27

**Authors:** Yaying Wu, Yujie Mou, Yu Zhang, Yu Han, Lin Lin, Yanan Huo, Yirui Zhu, Shuo Yang, Xiaodan Huang

**Affiliations:** Eye Center, The Second Affiliated Hospital, Zhejiang University School of Medicine, 88 Jiefang Road, Hangzhou 310009, China; 11918413@zju.edu.cn (Y.W.); 3140105584@zju.edu.cn (Y.M.); zhangyu2020@zju.edu.cn (Y.Z.); bella_hany@outlook.com (Y.H.); icylinlin@zju.edu.cn (L.L.); 3309035@zju.edu.cn (Y.H.); yiruizhu@zju.edu.cn (Y.Z.); 2317047@zju.edu.cn (S.Y.)

**Keywords:** intense pulsed light, dry eye disease, deproteinized calf blood extract eye drops, ocular pain

## Abstract

Purpose: To investigate the efficacy of intense pulsed light (IPL) combined with deproteinized calf blood extract (DCBE) eye drops for dry eye disease (DED) patients with nociceptive ocular pain. Methods: In this prospective, one-center, interventional study, 23 subjects with DED and ocular pain were treated with a combination of IPL and DCBE eye drops for four sessions at a four-week interval. Subjective and objective assessments on nociceptive pain and dry eye were examined and analyzed. Results: The visual analog scale (VAS), ocular surface disease index, ocular pain assessment survey (OPAS), patient health questionnaire-9 items, generalized anxiety disorder (GAD-7), Athens insomnia scale, corneal fluorescein staining score, meibomian gland secretion quality, and expressibility scores were significantly reduced after the treatment. Tear break-up time and Schirmer I test increased significantly. The brand density of corneal nerves and neuropeptide substance P also significantly increased. OPAS, GAD-7, meibomian gland secretion quality, and expressibility scores were essential factors affecting the VAS changes. Conclusions: IPL combined with DCBE drop therapy was effective for DED patients with ocular pain. With such treatment, both DED symptoms and the sensation of ocular pain may be improved.

## 1. Introduction

Dry eye disease (DED) is considered a complicated and multifactorial disorder of the ocular surface, with the prevalence ranging from 5 to 50% in individuals worldwide [1]. According to the International Dry Eye Workshop (2017), DED was defined as “a multifactorial disease of the ocular surface characterized by a loss of homeostasis of the tear film and accompanied by ocular symptoms, in which tear film instability and hyperosmolarity, ocular surface inflammation and damage, and neurosensory abnormalities play etiological roles” [2]. A heavy burden is brought on public finance by the prevalence of DED, with an estimated annual cost of USD 3.84 billion in the United States [3]. Several typical clinical symptoms are encompassed by DED, including the sensation of eye dryness, irritation, burning, soreness, and ocular pain [4,5,6].

Ocular pain is the third most critical symptom that needs attention at clinics [7]. Generally, pain can be divided into nociceptive, because of the nociceptive sensory neuron activities, and neuropathic, caused by the functional disturbance of neuroaxis [8]. The homeostasis of the ocular surface is broken by DED, damaging neuron nociceptors and peripheral nerves in the cornea and conjunctiva will result in nociceptive pain [9]. Furthmore, ocular pain is usually accompanied by irritation and photophobia, which negatively impact the quality of life. Meibomian gland dysfunction (MGD) is regarded as a sub-type of evaporative DED with meibomian oil deficiency. The prevalence of MGD is especially higher in Asian countries, ranging from 46.2% to 68.0% [10,11,12,13,14]. MGD induces higher evaporation which causes hyperosmolarity and inflammation on the ocular surface, even damage to corneal epithelium and nociceptors. Therefore, MGD has a close relationship with the generation of nociceptive ocular pain. 

DED has a variety of effective treatments, such as artificial tears, hot compresses, anti-inflammatory and immunosuppressant eye drops, autologous serum tears (AST), intense pulsed light (IPL), and vectored thermal pulsation [15]. Although neurosensory dysfunction has been proposed by several studies as an essential DED component, DED patients with ocular pain do not receive specific treatments for their symptoms in the clinical setting. IPL has traditionally been used to treat dermatological conditions and has been newly developed to alleviate the ocular discomfort and irritation associated with MGD [16,17]. A broad wavelength of light provided by the IPL, ranging from 500 to 1200 nm [18], is flashed upon the skin to trigger the coagulation of superficial blood vessels has been proven to be a reliable and effective therapy [19]. IPL was proved to have confirmed effects on reducing ocular inflammation, repairing corneal epithelium and tear film, increasing tear break-up time [16,17]; however, the efficacy of IPL for the treatment of nociceptive pain still remains unknown. Therefore, this study was designed to evaluate the efficacy of IPL treatment towards ocular pain.

In recent studies, AST was investigated to be a highly effective management for severe corneal pain without obvious clinical signs. AST can increase corneal nerve density, restore injured neurons and contribute to the growth of sensory and sympathetic neurons. In our previous study, deproteinized calf blood extract (DCBE) was an effective topical treatment for relieving ocular pain raised by DED [20]. DCBE is a protein-free hemodialysate mainly containing various free amino acids and low-molecular peptides and oligosaccharides of blood serum obtained from veal calves [21,22,23]. DCBE eye drops have similar characteristics, functioning as AST and having considerable repair effects for the tear film. 

Concerning the potential function of repairing tear film and corneal nerve, reducing neuroreceptors irritation and ameliorating corneal pain, we combined IPL therapy and DCBE eye drops in this study to investigate whether this application is an effective treatment for patients diagnosed with DED with specific ocular pain.

## 2. Methods

### 2.1. Subjects

We recruited 23 patients aged 22–75 years, including five males and 18 females. This prospective study was conducted at the Second Affiliated Hospital of Zhejiang University School of Medicine between September 2019 and July 2020. The tenets of the Declaration of Helsinki were followed in this study, and it was approved by the Ethics Committee of the Second Affiliated Hospital of Zhejiang University School of Medicine. Written informed consent was obtained from each participant after the nature, and the possible results of the study were explained (Trial registration number: ChiCTR1900022911). Patients with tear break-up time (TBUT) under 10 s and obvious dry-eye-related hyperesthesia symptoms (light sensitivity or ocular pain) which can be relieved after surface anesthesia were diagnosed as DED patients with ocular pain and were recruited for the study. The exclusion criteria included: (1) eye allergies, (2) infectious conjunctivitis, (3) corneal ulcer, (4) eyelid infection, (5) severe blepharitis, (6) palsy, (7) valgus or other ocular lesions, (8) severe systemic disease, and (9) history of wearing contact lenses in the preceding 30 days.

The patients were treated with IPL four times at a four-week interval. Patients were also asked to use DCBE eye drops (Qixing Pharmaceutical, Shenyang, China) four times a day after each consecutive IPL treatment for 16 weeks. All patients were followed up every four weeks after each IPL treatment.

### 2.2. Questionnaires

#### 2.2.1. Visual Analog Scale (VAS) of Ocular Pain

The subjects were asked to score their own general ocular pain perceptions on a visual analog scale scored from 0 to 10. Self-reported ocular pain was scored as no pain (0) up to the severest pain (10) ever suffered.

#### 2.2.2. Ocular Surface Disease Index (OSDI)

The OSDI is recommended by TFO DEWS II for diagnosing DED, and its reliability and efficacy have been confirmed. The OSDI contains 12 different questions and scores 0–4, representing the slight to a severe degree. After aggregating the scores, the total score was 0–100. An OSDI score over 13 was diagnosed as symptomatic DED [24].

#### 2.2.3. Ocular Pain Assessment Survey (OPAS)

OPAS was designed as an initial 32-question, 8-domain, ocular pain questionnaire using numerical rating scales to assess eye pain intensity (worse eye) in the past 24 h and two weeks (worst, least, and average pain intensity), frequency of eye and non-eye pain (past 24 h and two weeks), non-eye pain intensity (past 24 h and two weeks), impact on quality of life (QoL), pre-occupation with eye and non-eye pain, aggravating factors, associated factors, and symptomatic relief. Questions on the frequency of pain were included to assess the discriminant validity of the OPAS since, hypothetically, any observable relationship with the frequency of ocular pain should not be demonstrated by the intensity of ocular pain. Each test question was scored on a scale of 0–10, or 0–100%, with increments of 1% or 10% units, respectively. The questions were divided into sections for analysis: questions 4–9 pertained to the intensity of ocular pain; questions 10–12, non-ocular pain; questions 13–19, QoL; questions 20–21, aggravating factors; and questions 22–25, associated factors. After excluding the section on symptomatic relief, only questions 4–25 were analyzed in this study [25].

#### 2.2.4. Patient Health Questionnaire-9 Items (PHQ-9)

In this survey of nine questions, patients were asked about their frequency of being bothered by problems (e.g., displeasure in doing things, feeling down, trouble concentrating, feeling bad about oneself, and suicidal ideation) over the past two weeks. A cumulative score from 0 to 27 was assigned, with scores related to depression severity (minimal depression to severe depression). For this study, we conservatively dichotomized depression scores into those with the lowest total scores (scores 0–9, minimal depression, and mild depression) as “not depressed,” while those with higher total scores (scores 10–27, moderate depression to severe depression) were categorized as “depressed” [26].

#### 2.2.5. Generalized Anxiety Disorder (GAD-7)

The 7-item generalized anxiety disorder (GAD-7) scale is a validated, reliable, and efficient measurement tool for GAD screening and assessing its severity in clinical practice and research. In the GAD-7 scale, four levels of anxiety severity are classified: none (0–4), mild (5–9), moderate (10–14), and severe (15–21) [27].

#### 2.2.6. Athens Insomnia Scale (AIS)

The AIS is a self-assessment psychometric instrument designed for quantifying sleep difficulty based on the ICD-10 criteria. It consists of eight items: the first five pertain to sleep induction, awakenings during the night, final awakening, total sleep duration, and sleep quality. The last three are related to well-being, functioning capacity, and sleepiness during the day. They are based on a 0–3 scale, in which three designates adverse outcomes. Total AIS scores range from 0 to 24 points. A minimum total score of six points is an indication of an insomnia diagnosis [28].

### 2.3. DED Examination

#### 2.3.1. Corneal Fluorescence Staining (CFS) Score

The cornea was divided equally into four quadrants. The scoring was recorded after staining as follows: (0) no dye point, (1) 1–30 points, (2) more than 30 dye spots but no fusion, and (3) entirely stained with fusion. A total score in each eye ranges from 0 to 12. Scores over 1 point is regarded as defects of corneal epithelium.

#### 2.3.2. TBUT Test

Tear film instability was assessed using TBUT measurements. First, fluorescein was added to the tear film, and the patients were asked to blink entirely several times to ensure that it was uniformly distributed across the ocular surface. The time between the last blink and the first dry spot on the tear film was measured. TBUT under 10 s was a golden diagnostic criteria for DED.

#### 2.3.3. Schirmer I Test (SIT)

A filter paper strip (35 mm × 5 mm) was put into the lower eyelid conjunctival sac after applying topical anesthesia, and the lengths of the wet parts were recorded after 5 min. Lengths shorter than 10 mm were considered abnormal.

#### 2.3.4. Meibomian Gland Expressibility Score

The meibomian gland evaluator (MGE, TearScience, Inc., Morrisville, NC, USA) was used to simulate the constant gentle pressure (1.2 g/mm^2^) on the lower eyelid of the participant, similar to the pressure of an average blink, and then the meibomian gland orifices and secretions were observed through a slit-lamp microscope. Three parts of the upper and lower eyelids were evaluated using the meibomian glands expressibility score in the temporal, central, and nasal parts of the lower eyelid. Each part contained five glands. If the five glands in the center of the meibomian glands were normal, zero points were assigned. One point was assigned if either one or two glands without secretions were mildly abnormal. Two points were assigned if only three or four glands discharged secretions with moderate abnormalities. Finally, three points were assigned if there were five glands without secretions for severe abnormalities. The meibomian gland expressibility score for moderate and severe abnormalities suggested the possibility of dry eye, with a total score of 18 points.

#### 2.3.5. Meibomian Gland Secretion Quality Score

The MGE was placed in the lower eyelid 2 mm from the root of the eyelashes, 45° upward, and maintained for 10–15 s. Five glands were evaluated in the temporal, central, and nasal parts of the lower eyelid, with 15 glands in total. The secretion quality assessment of each of the 15 glands in the lower eyelid was scored 0–3. Such scoring corresponded to no secretion (grade 3), inspissated secretion (grade 2), cloudy secretion (grade 1), and clear secretion (grade 0), with total scores of 45 points. 

#### 2.3.6. In Vivo Confocal Microscopy (IVCM)

IVCM evaluation of the central cornea was performed using Heidelberg retinal tomography (HRT) III with an anterior segment module (Heidelberg Engineering GmbH, Heidelberg, Germany). This was followed by the installation of a drop of proparacaine hydrochloride eye drops (s.a. ALCON-COUVREUR n.v.) into the conjunctival fornix. The objective lens was covered with a disposable polymethacrylate sterile cap, and Viscotears was applied as the coupling agent. Participants were requested to fix their gaze on a central target to allow for full-thickness scanning of the central cornea in 2 μm increments using the section mode setting of the tomograph. Parameters of corneal nerve fiber length (CNFL), fiber density (CNFL), brand nerve density (CNBD), fiber tortuosity (CNFT), abnormal corneal nerve fiber, and dendritic cells were evaluated.

#### 2.3.7. Tears Collection

All study participants had tear secretion collected at about 14:00 in a quiet room with weak light without surface anesthetic. Tear samples were collected for each eye with 200 μL normal saline and immediately placed in an EP tube and stored in a deep cryogenic refrigerator at −80 °C to avoid repeated freezing and thawing.

#### 2.3.8. Luminex Assay

The cytokine levels of the tear samples were performed in triplicate. The cytokines included TNF-α, IL-6, and IL-8. The concentration values were obtained from the mean fluorescent intensity (MFI) using Luminex200 IS V2.1 software. Standard curves were generated from the reference cytokine gradient concentrations. The concentrations of these cytokines in the tear samples were calculated from standard curves. 

#### 2.3.9. ELISA Assay

The CGRP and substance P levels in the tear samples were determined using the ELISA assay. Analyses were performed in triplicate according to the manufacturer’s instructions for each ELISA kit.

### 2.4. Statistics

All statistical analyses were performed using SPSS software (version 25.0, IBM, New York, NY, USA). The Kolmogorov–Smirnov test was used to identify the normal distribution of continuous variables. A *p*-value of < 0.05 was considered statistically significant. In the raw analyses, *t*-tests were performed for normally distributed continuous variables, including the OSDI, TBUT, SIT, meibomian gland expressibility, secretion quality scores, CNFL, and the concentration of cytokines and neuropeptides in tear samples. Kruskal–Wallis tests were conducted for ordinal outcomes, including VAS, OPAS, PHQ-9, GAD-7, AIS, and CFS scale. The analysis was double-sided, and a *p*-value of <0.05 was considered statistically significant.

## 3. Results

### 3.1. Baseline Characteristics

All patients were diagnosed with DED with ocular pain in both eyes. Fourteen of them announced a bilateral headache of facial pain, while eight patients reported no headache or facial pain. Only one patient had a headache or facial pain on his right side. Regarding the psychological assessments, 10 patients had insomnia, five had depression, and five had anxiety. Nine patients had a history of ocular trauma or surgery, and nine patients had systemic diseases. Complete demographic information is shown in Table 1. 

### 3.2. Improvements in Psychological Symptoms of DED Patients with Ocular Pain Using IPL Combined with DCBE Eye Drops

Changes in subjective symptoms after every therapy session are shown in Figure 1. The OSDI and VAS scores decreased significantly after the first session of IPL combined with DCBE eye drops and reached their minimum at the 16th week. The OSDI and VAS scores were significantly lower than the baseline information at all visits. Considering the OPAS assessment at 24 h and two weeks, OPAS decreased significantly after the first treatment and reached its minimum at the 16th week. OPAS assessment of QoL, non-eye pain, and associated factors reduced significantly after the second session in the fourth week and reached their minimum at the 16th week. The aggravating factors of OPAS only had a significant decrease in the last session of the 16th week. PHQ-9, GAD-7, and AIS declined significantly in the 12th week, although no significant changes occurred in the fourth and eighth weeks.

### 3.3. Dry-Eye-Related Signs of DED Patients with Ocular Pain Improved by IPL Combined with DCBE Eye Drops

Improvements in dry eye examinations that occurred after four IPL treatment sessions combined with DCBE eye drops are shown in Figure 2. The CFS decreased significantly after the first therapy session and reached its minimum at the 16th week. Similarly, TBUT increased since the first treatment and was significantly increased at every follow-up. The tear function examined in the Schirmer I test improved significantly in the 12th and 16th weeks. Regarding the meibomian gland function evaluation, the meibum expressibility scores decreased significantly after the first therapy. The meibum secretion quality scores began to decline significantly after the third therapy. The function of the meibomian glands improved significantly after four combined therapy sessions.

### 3.4. Corneal Nerves Brand Density of DED Patients with Ocular Pain Improved with IPL Combined with DCBE

The ocular pain sensation could be directly affected by corneal nerve characteristics. The observed changes are displayed in Figure 3. We evaluated seven different items of corneal nerves by IVCM and found only a significant difference in brand density. The brand density of corneal nerves was significantly higher after four combined therapy sessions than during pre-treatment. No significant difference was found in the CNFL, CNFD, CNFT, ACNF, or dendritic cells.

### 3.5. Substance P in DED Patients with Ocular Pain Increased with IPL Combined with DCBE Eye Drops

Topical inflammation and pain may be caused by the accumulation of related inflammatory cytokines. We examined five relevant cytokines before the treatment, after the second treatment in the eighth week, and after the fourth treatment in the sixteenth week. The results are shown in Figure 4. According to our results, there was a significant increase in substance P after the fourth therapy. CGRP, IL-6, IL-8, and TNF-α did not significantly change during the therapeutic sessions.

### 3.6. VAS Scores Were Affected by Several Factors after Combined IPL and DCBE Eye Drops Treatment of DED Patients with Ocular Pain

The VAS is considered one of the simplest and most sensitive tools for reporting degrees of pain and discomfort. In our study, we divided patients into two groups to find which factors affected the changes in VAS scores. The groups considered were the normal-response group, in which VAS scores decreased less than 50%, and the sensitive-response group, in which VAS scores decreased at least 50%. The comparative results are shown in Table 2. Before the combined IPL and DCBE eye drops therapy, no significant differences between the normal-response and sensitive-response groups. However, after four combined therapy sessions, OPAS scores after 24 h and non-eye pain in the sensitive-response group significantly decreased compared with the normal-response group. The anxiety analytical scale GAD-7 also significantly differed between the two groups. The sensitive-response group had lower meibum expressibility and secretion quality scores than the normal-response group, indicating that meibomian gland function was an important factor that affected the VAS scores.

## 4. Discussion

Multiple studies have been designed to investigate effective treatments for DED. However, few studies have systematically focused on the efficacy of these treatments specifically for the ocular pain symptoms of DED. Our study was the first to investigate the effectiveness of a combination treatment of IPL and DCBE eye drops on DED accompanied by ocular pain. In our prospective study, we found that IPL combined with DCBE eye drops for four sessions was an effective treatment to relieve ocular pain in DED patients. Subjective symptoms, DED signs, and laboratory markers improved after the novel therapy. The VAS, OSDI, OPAS, PHQ-9, GAD-7, and AIS scores decreased significantly during the research. The CFS, TBUT, Schirmer I test, meibum expressibility, and secretion quality improved. Furthermore, the corneal nerve brand density was repaired by the combined treatment of IPL with DCBE eye drops, with which the levels of substance P were also improved, according to our results.

The homeostasis of the tear film is broken by DED, inducing ocular hyperosmolarity and inflammation [29]. Topical pathological changes can damage and sensitize the nociceptors and peripheral nerves in the cornea and conjunctiva, leading to hyperalgesia or allodynia in DED patients [30]. Long-term stimulation without timely treatment will undermine neuroimmune communication and move into chronic ocular, even systemic pain [5]. Although ocular pain has been noted and emphasized by TFOS DEWS II pain and sensation reports and many ophthalmologists [9], specific therapy targeting ocular pain has rarely been reported. In this study, we proposed using IPL combined with DCBE eye drops to relieve ocular pain and achieve a positive outcome.

In our study, we showed that after four treatment sessions, the CFS scores were significantly reduced. Meanwhile, TBUT and Schirmer I test results were significantly elevated in all therapy stages. These results are indications that, with the combined treatment of IPL with DCBE eye drops, the tear film could be stabilized, building a healthy ocular micro-environment. TBUT increased an average of 3.65 times (2.03–7.40 s), and there was an average of 62.81% (8.04–13.09 mm) in the Schirmer I test results compared with the baseline. This represents a meaningful clinical improvement and a statistically significant improvement in tear film stability and tear volume. In our study, we found that meibum expressibility and secretion quality were improved by IPL combined with DCBE eye drops, especially during the last two visits. The meibum in the glands may be melted by a high IPL temperature, accelerating its discharge and circulation. These results are in accordance with several previous studies [17,18,31,32].

With the IVCM examination, the brand density of the corneal nerves was enhanced after treatment. At the same time, the length, tortuosity, and relevant cells had no differences. Time and frequency may result in this effect. We presumed that if the treatment sessions were prolonged in our study, the response of the corneal nerves might have been significantly increased. For inflammatory cytokines, only substance P was found to be significantly increased. Although CGRP, IL-6, IL-8, and TNF-α had a decreasing tendency, the analytical results were not different. This was in accordance with the study by Liu et al. [16]. Substance P is a sensitive neuropeptide distributed in the nerve fibers of corneal epithelial cells and stromal cells. Substance P has been reported to have analgesic, nerve repairment, and vasodilation abilities [33]. With our results, we suggest that substance P might respond faster than other related cytokines.

After four combined treatment sessions, the subjective and sensory responses seemed more sensitive and evident. Regarding topical eye sensation, the VAS and OSDI scores decreased at every stage of the treatment and were reduced to nearly one third of the baseline. Six OPAS assessment items decreased after the final therapy, which indicated that treatment with IPL combined with DCBE eye drops could alleviate the lasting time and degree of ocular pain and promote non-ocular feelings and QoL. Psychological improvements were also demonstrated by the PHQ-9, GAD-7, and AIS reduced scores, implying that the systematic feelings of anxiety, depression, and insomnia were alleviated by the combined IPL and DCBE eye drops treatments.

We next divided the patients into two groups based on the results of the VAS changes. The overall therapeutic effects of the combined treatment were represented by the VAS. The two groups did not differ in all parameters before treatment. However, after four combined therapy sessions, the OPAS-24H and non-eye pain, GAD-7, meibomian gland expressibility, and secretion quality scores decreased significantly. This is an indication that therapeutic effects might be affected by the degree of pain, systemic anxiety, and meibomian gland function. Apparent relief in anxiety control and improvement of meibomian glands can be expected to produce better curative outcomes.

There are several potential mechanisms for alleviating subjective ocular symptoms and objective signs by combining IPL and DCBE eye drops in DED patients with ocular pain. IPL has been widely reported to impact MGD positively [34], and the patients involved in our study had higher meibomian expressibility and secretion quality scores. The theoretical reason mainly focuses on the anti-inflammation and stability of the tear film. First, the high IPL temperature applied to the periocular skin can warm the meibomian glands and melt the meibum [35]. The secretion of old and generation of new meibum are promoted by meibum melting, contributing to tear film stability, osmolarity reduction, and inflammation suppression [36]. Second, fibroblasts may be activated by broad-spectrum IPL, enhancing collagen synthesis and inhibiting neovascularization of the cornea and eyelid [37]. Third, bacteria in the eyelid margin may be immediately reduced by IPL treatment, thereby attenuating inflammation [35]. Inflammatory cytokines, such as IL-6, IL-8, and TNF-α, were prevented from a stable and healthy tear film and the loss of irritation by inflammation. This could be responsible for reducing the sensitivity of nociceptors, inactivating the pain receptors, protecting the corneal nerves, and thereby relieving ocular pain. Although it is still unknown whether the corneal nerves are directly affected by IPL, a possible close relationship between IPL treatment and the repair of the corneal nerves was implied in our study.

In previous studies, AST was proven to be beneficial to patients with corneal neuropathy-induced photoallodynia or severe corneal pain by promoting nerve regeneration and repairing nerve topography [38]. Nevertheless, AST is prevented from extensive use in China because of its high cost and strict preservation requirements. DCBE eye drops are a proper alternative. DCBE eye drops consist of adequate free amino acids, low-molecular peptides, and oligosaccharides [22]. Oxygen and glucose transportation in a hypoxic environment is allowed by DCBE eye drops, providing adequate ATP for aerobic glycolysis and oxidative phosphorylation [23]. The repair and regeneration of corneal peripheral nerves may be accelerated by frequent DCBE eye drops, reducing the release of inflammatory factors by nerve endings. Ocular pain is also relieved by DCBE by offering nutrients and oxygen to the ocular surface. By combining IPL and DCBE eye drops, the ocular environment is stabilized, suppressing topical inflammation [21] and promoting nerve repairment and regeneration. Therefore, reliable and effective treatment for DED patients is required, specifically for those with ocular pain.

Compared to traditional treatments for DED patients with apparent ocular pain, IPL combined with DCBE eye drops has several advantages. To relieve nociceptive pain induced by eye dryness, anti-inflammatory and anti-depressant drugs are adopted [39,40]. However, these drugs can solely target on symptoms and may have severe side effects and addictions. Topical nerve blocking treatments, such as lidocaine and corticosteroid, can effectively reduce painful perception [41], but do not contribute to the cause of DED. IPL combined with DCBE eye drops can not only relieve nociceptive painful symptoms and repair the ocular inflammatory environment, but also improve curative effects on DED aetiology.

Our study has some limitations. First, the number of enrolled subjects may not be sufficient to determine guidelines for the combined treatment of IPL with DCBE. Furthermore, we need a more significant number of participants to validate the efficacy and conclusion of the current study. Second, our study failed to involve a control group in this experiment, and a randomized and controlled, long-term, multi-center study is essential for further research. The IVCM examination and cytokines did not have apparent changes, and long-term research might be more helpful. Third, due to the apparatus limits, our study failed to evaluate the possible changes of ocular osmolarity, as hyperosmolarity is considered as an essential cause of DED and ocular pain [29]. IPL was proved to reduce the tear osmolarity after 4 to 6 sessions [42]; therefore, the combined treatment may also relieve nociceptive pain through decreasing tear osmolarity. This may need some further research.

In conclusion, our study revealed that IPL combined with DCBE eye drops is an effective and reliable treatment for managing DED patients with ocular pain. The combined treatment had positive effects on both dry eye symptoms and the sensation of ocular pain. This may provide a novel guideline for clinical DED treatment.

## Figures and Tables

**Figure 1 jcm-11-01312-f001:**
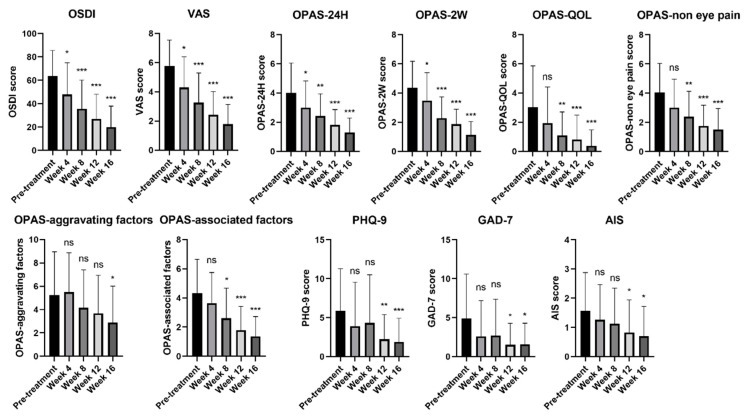
Changes of DED symptoms and ocular pain before and after treatment of IPL combined with DCBE eye drops. *** *p* < 0.001, ** *p* < 0.01, * *p* < 0.05, ns: no significance. OSDI: Ocular Surface Disease Index, VAS: Visual Analog Scale, OPAS: Ocular Pain Assessment Survey, PHQ-9: Patient Health Questionnaire-9 Items, GAD-7: Generalized Anxiety Disorder 7, AIS: Athens Insomnia Scale.

**Figure 2 jcm-11-01312-f002:**
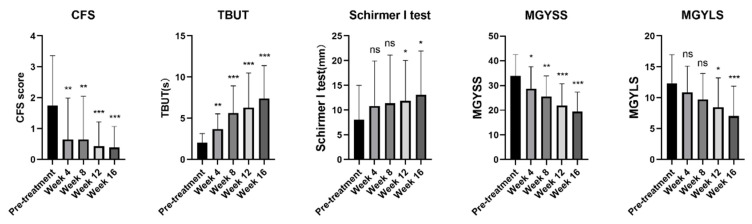
Changes of signs and examinations before and after treatment of IPL combined with DCBE eye drops. *** *p* < 0.001, ** *p* < 0.01, * *p* < 0.05, ns: no significance. CFS: Corneal Fluorescein Staining, TBUT: Tear Break-up Time, MGYSS: Meibomian Gland Secretion Quality Score, MGYLS: Meibomian Gland Expressibility Score.

**Figure 3 jcm-11-01312-f003:**
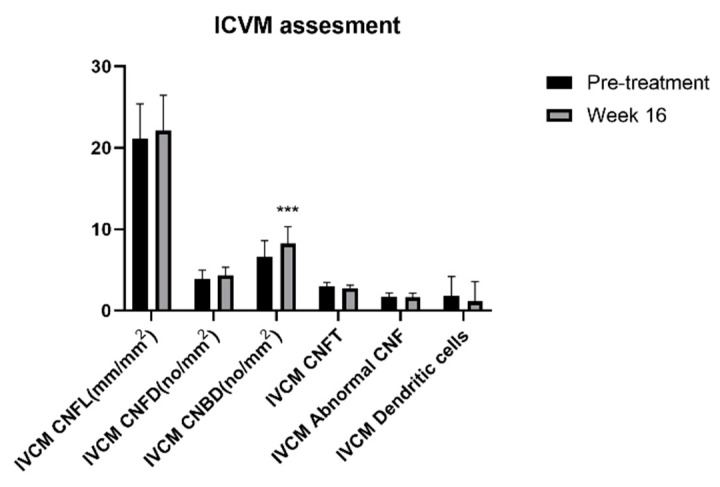
Changes of IVCM assessments before and after treatment of IPL combined with DCBE eye drops. *** *p* < 0.001. IVCM: In Vivo Confocal Microscopy, CNFL: Corneal Nerve Fiber Length, CNFL: Corneal Nerve Fiber Density, CNBD: Corneal Nerve Brand Density, CNFT: Corneal Nerve Fiber Tortuosity, CNF: Corneal Nerve Fiber.

**Figure 4 jcm-11-01312-f004:**
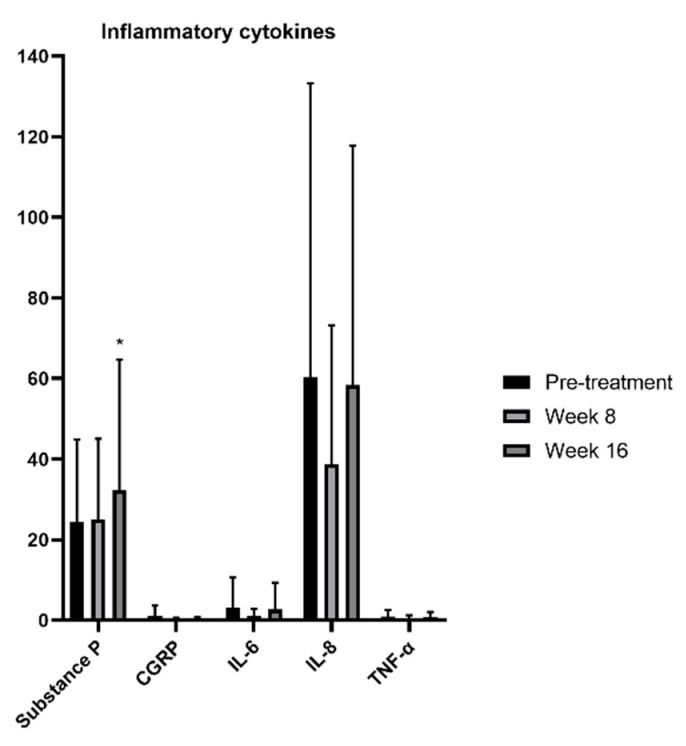
Changes of inflammatory cytokines before and after treatment of IPL combined with DCBE eye drops. * *p* < 0.05.

**Table 1 jcm-11-01312-t001:** Demographics of subjects.

	Subject (*n* = 23)
Age	46.83 ± 14.77
Sex (male:female)	5:18
Years after diagnosis	1.91 ± 1.12
Double eye pain	23
No headache or facial pain	8
Right side headache or facial pain (with right eye pain)	1
Left side headache or facial pain (with left eye pain)	0
Bilateral headache or facial pain	14
Insomnia	10
Depression	5
Anxiety	5
Ocular disease or with history of ocular trauma or surgery	9
Systemic diseases	9
Sjogren Syndrom	1

Data were presented with mean ± SD.

**Table 2 jcm-11-01312-t002:** Changes of factors affect VAS scores after IPL combined DCBE eye drops treatment in DED patients with ocular pain.

	Pre-Treatment			Week 16		
	Normal-Response Group	Sensitive-Response Group	*p*	Normal-Response Group	Sensitive-Response Group	*p*
VAS	4.92 ± 1.78	6.73 ± 1.19		3.13 ± 1.37	1.18 ± 1.08	
OSDI	65.82 ± 22.58	61.02 ± 22.10	0.689	22.31 ± 18.06	17.58 ± 18.30	0.478
OPAS						
OPAS-24H	3.78 ± 2.26	4.27 ± 1.84	0.174	1.83 ± 0.96	0.7 ± 0.66	0.004 **
OPAS-2W	3.94 ± 2.3	4.79 ± 1.02	0.169	1.92 ± 1.14	1.82 ± 0.92	0.701
OPAS QOL	3.22 ± 2.99	2.82 ± 2.80	0.709	0.56 ± 1.45	0.21 ± 0.40	0.968
OPAS non-eye pain	3.75 ± 1.51	4.4 ± 2.42	0.518	2.01 ± 1.42	0.97 ± 1.31	0.026 *
OPAS aggravating factors	5.36 ± 3.3	5.14 ± 4.25	0.802	3.17 ± 2.84	2.59 ± 3.52	0.474
OPAS associated factors	4.15 ± 2.53	4.55 ± 2.14	0.779	1.38 ± 0.86	1.36 ± 1.78	0.223
PHQ-9	7.00 ± 6.34	4.64 ± 4.13	0.353	2.58 ± 3.90	1.18 ± 1.47	0.548
GAD-7	4.75 ± 6.15	5.00 ± 5.59	0.900	2.75 ± 3.36	0.27 ± 0.65	0.011 *
AIS	1.58 ± 1.24	1.55 ± 1.44	0.948	0.83 ± 1.03	0.55 ± 1.04	0.361
CFS	2.08 ± 2.19	1.36 ± 1.03	0.743	0.58 ± 0.9	0.27 ± 0.65	0.270
TBUT (s)	1.76 ± 0.81	2.33 ± 1.38	0.389	7.80 ± 4.42	6.97 ± 3.63	0.854
Schirmer I test (mm)	9.50 ± 7.63	6.45 ± 6.11	0.138	14.08 ± 8.50	12.00 ± 9.53	0.439
MGYSS	33.92 ± 6.56	29.55 ± 8.57	0.109	21.92 ± 5.73	14.82 ± 9.24	0.024 *
MGYLS	12.58 ± 5.23	9.91 ± 4.74	0.115	9.42 ± 4.19	5 ± 3.82	0.019 *

Data were presented with mean ± SD. ** *p* < 0.01, * *p* < 0.05. OSDI: Ocular Surface Disease Index, VAS: Visual Analog Scale, OPAS: Ocular Pain Assessment Survey, PHQ-9: Patient Health Questionnaire-9 Items, GAD-7: Generalized Anxiety Disorder 7, AIS: Athens Insomnia Scale. CFS: Corneal Fluorescein Staining, TBUT: Tear Break-up Time, MGYSS: Meibomian Gland Secretion Quality Score, MGYLS: Meibomian Gland Expressibility Score.

## Data Availability

The data presented in this study are available on request from the corresponding author. The data are not publicly available due to the Ethical Review Board has not approved the public availability of these data.

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
