# Peer review of "Efficacy of Intense Pulsed Light Combined Blood Extract Eye Drops for Treatment of Nociceptive Pain in Dry Eye Patients"

_jcm, 2022, doi:10.3390/jcm11051312_

Round 1
Reviewer 1 Report
This is a cross sectional observational study on the effect of IPL and DCBE eye drops for the treatment of DED. Off-the-bat, the study is done well and the conclusions are not over exaggerated and have meaningful data coming out of it. However, it is hard to assess the true effect of these treatments without a placebo. However, I do not think it will be possible to repeat these with a placebo and the results have to be taken with a grain of salt for this reason.
It would be helpful to understand the contents of the DCBE if possible. I request the authors to submit that to the journal.
I request the authors to discuss a little more regarding how this treatment compares to some other approved treatments available in the clinic
Author Response
Response to Reviewer 1:
Dear Reviewer:
We appreciate for your comments concerning our manuscript entitled “Efficacy of Intense Pulsed Light Combined Blood Extract Eye Drops for Treatment of Nociceptive Pain in Dry Eye Patient”. We deeply appreciate the time and detail provided by you and editor and we have incorporated the suggested changes into the manuscript to the best of our ability. Those comments are all valuable and very helpful for revising and improving our paper. Based on these suggestions, we have made extensive modifications on the revised manuscript. Revised portion was marked with “track changes” in the revised manuscript.
According to your first comment, it is a real limitation that our study failed to involve a control group in this experiment. Perhaps in the future, a randomized and controlled, long-term, multi-center study will be performed.
According to your second comment, DCBE eye drops mainly contain a variety of free amino acids, low molecular peptides, and oligosaccharides, containing a broad spectrum of low molecular components of cellular mass and blood serum obtained from veal calves. DCBE has been widely used in a variety of ocular surface diseases in recent years, showing good improvement in patients suffering from ocular surface symptoms. We have added these statements in the INTRODUCTION part (line72-75) to give a clearer illustration of DCBE.
According to your third comment, relevant discussions on other approved treatments were added to the DISCUSSION part (line414-422). We illustrated several traditional treatment for DED with nociceptive pain and made comparisons between our treatment and traditional treatments.
We sincerely appreciated your helpful comments and suggestions and hope that our revisions could meet your demands.
Reviewer 2 Report
It is a well done study on the therapeutic effects of a new combination of dry eye treatments. The authors used a wide spectrum of questionnaires, examinations and laboratory assessments to follow the changes during the treatment. The results show promising treatment outcomes. The manuscript needs minor corrections.
Detailed comments:
Methods
Please, detail the diagnostic criteria which were used for DED diagnosis and for ocular pain.
Among exclusion criteria (2) and (4) are not well defined as DED and MGD are usually including conjunctival and eye-lid inflammations. Even ocular surface inflammation is a diagnostic criterion for DED, as also the authors mentioned it in the 6 row of their Introduction and in the 1st sentence in the 2nd paragraph of the Discussion. Please, provide more details to avoid confusion.
In some questionnaire and examinations, the authors provided categorisations but not in all. Please, provide normal-abnormal limits for all of them.
2.3.4. and 2.3.5. Both meibomian gland secretion examination needs expression on the glands. What is the opinion of the authors about the possible influence of the first performed invasive test maneuver to the results of the secondly performed other test?
What was the order of the different examinations and tear film sampling? What was the time interval between the tear collections and the tear film and meibomian gland invasive examinations?
Results
The number, age and sex data of the patients need to be in the Methods section.
Figure 1 and Table 2 contains the same information. Possibly there is no need for double (redundant) presentation of the results?
Usually the Tables and Figures need to be understandable if a reader sees only them. Therefore, there is a need to describe the meaning of the abbreviations in the figure legends and in the tables.
Discussion
It is a limitation of the present study that tear osmolarity measurement was not included and there is no data on the effects of the applied therapy on tear film osmolarity. As some earlier studies found that ocular pain is connected to the level of tear hyperosmolarity, please, address the topic of the possible role of tear hyperosmolarity and ocular pain.
Author Response
Response to Reviewer 2:
Dear Reviewer:
We appreciate for your comments concerning our manuscript entitled “Efficacy of Intense Pulsed Light Combined Blood Extract Eye Drops for Treatment of Nociceptive Pain in Dry Eye Patient”. We deeply appreciate the time and detail provided by you and editor and we have incorporated the suggested changes into the manuscript to the best of our ability. Those comments are all valuable and very helpful for revising and improving our paper. Based on these suggestions, we have made extensive modifications on the revised manuscript. Revised portion was marked with “track changes” in the revised manuscript.
According to your comments on the METHOD part, we detailed the diagnostic criteria for our study: patients with tear break-up time (TBUT) under 10 seconds and obvious dry-eye-related hyperesthesia symptoms (light sensitivity or ocular pain) which can be relieved after surface anesthesia were diagnosed as DED patients with ocular pain and were recruited for the study. (line 89-92)
We felt sorry for not clearly illustrating the exclusion criterion (2) and (4). We use “infectious conjunctivitis” to replace criteria (2) and “eyelid infection” to replace criteria (4). (line 93)
For the normal-abnormal limits of questionnaires and examinations, we added the cut-off point of CFS and TBUT measurements. VAS and OPAS questionnaires are designed to evaluate the degree of pain and quality of life, therefore, they do not demonstrate a normal-abnormal limits. (line 153, 158-159)
For 2.3.4. and 2.3.5, we assessed the meibomian gland expressibility and quality once at the same time by using the meibomian gland evaluator (MGE), because the number and area of meibomian glands were the same in these two assessments. (line 165-168)
For the order of our experiment, subjective questionnaires and objective ocular measurements for detecting dry eye and nociceptive pain were adopted from the most non-invasive to the most invasive:, subjective questionnaires, fluorescein tear break-up time (FTBUT), corneal fluorescein staining (CFS), , meibomian gland expressibility and quality, tear collectionSchirmer I test and IVCM. The time interval between tear collection and meibomian gland evaluation was 30 minutes to reduce the irritation of tear collection.
According to your comments on the RESULT part, the number, age and sex data of the patients were moved to the METHOD section.(line 82) Also, we added the meanings of abbreviations in all tables and figures.
As you suggested, Figure 1 and Table 2 demonstrated the redundant information, and Table 2 also have the same information with Figure 2, 3 and 4. So we decided to delete Table 2 to make our manuscript more concise.
According to your comments on the DISCUSSION part, we added more discussion on the relationship between hyperosmolarity and ocular pain. Due to the apparatus limits, our study failed to evaluate the possible changes of ocular osmolarity, as hyperosmolarity is considered as an essential cause of DED and ocular pain. IPL was proved to reduce the tear osmolarity after 4 to 6 sessions, therefore, the combined treatment may also relieve nociceptive pain through decreasing tear osmolarity. This may need some further researches. (line 429-433)
We sincerely appreciated your helpful comments and suggestions and hope that our revisions could meet your demands.